# communications
# engineering

# A deep learning-based predictive simulator for the optimization of ultrashort pulse laser drilling

Kohei Shimahara [1], Shuntaro Tani [1], Haruyuki Sakurai [1] & Yohei Kobayashi [1✉]

Ultrashort pulse laser drilling is a promising method for the fabrication of microchannels in dielectric materials. Due to the complexity of the process, there is a strong demand for numerical models (simulators) that can predict structures produced under specific processing conditions in order to rapidly find optimal processing parameters. However, the validity of conventional laser drilling simulators for dielectrics has been confined to a range of strict interpolations of the data used during the construction of the model, and thus, their usefulness is limited. Here, we demonstrate simulator-based optimization for ultrashort pulse laser drilling in dielectrics based on an iterative deep neural network which is trained to predict microchannel structure after a small number of irradiated pulses. Our approach predicts the development of hole shapes over a wide variety of conditions and allowed the discovery of 20% more energy efficient processing strategies than in the initial experimental data. More broadly, our approach can address realistic problems in laser processing considering a variety of processing parameters, and thus enabling improved performance of next-generation smart laser processing systems.

[1] The Institute for Solid State Physics, The University of Tokyo, Kashiwa, Chiba, Japan. ✉email: yohei@issp.u-tokyo.ac.jp

As the manufacturing industry evolves towards a paradigm of mass customization[1,2], there is a rising demand for cyber-physical systems[3,4] that can efficiently manufacture tailored products for individual customers. Laser processing is a candidate component for such manufacturing systems due to its precision, versatility, and compatibility with computer-aided control[5,6].

A key concept of cyber-physical systems is simulator-based optimization, which is the search for an optimal processing condition to produce a target design using a simulator. In this paper, we will refer to a simulator as a numerical model that can predict a structure produced under a given processing condition. The advantage of simulator-based optimization is that one can perform an optimal search on a computer without the need for trial-and-error in physical space. The utility of a simulator for optimization is directly linked to its range of validity, which provides hard bounds to the possible search space. Such limits often correspond to the range of the experimental data used for tuning the model[7–10]. In laser processing, expanding this range of validity and hence, creating a truly useful simulator, remains challenging due to the difficulty of constructing a model able to encompass the enormous parameter space available for laser processing.

Notwithstanding, extensive studies have been performed to construct models for laser processing. The approaches taken in these studies can be roughly categorized as microscopic or phenomenological. The microscopic approach models laser processing from the bottom-up from basic physical equations, and has helped elucidate fundamental processes such as electronic excitation and atomic motions;[11–17] however, a model that can predict structures on a practical scale has yet to be achieved. This is because laser processing is the accumulation of numerous phenomena distributed over a vast spatiotemporal scale, many of which are still under intense debate[18–20]. Therefore, the microscopic approach is not yet applicable for simulator-based optimization. The phenomenological approach, on the other hand, focuses on creating simplified models of ablation processes tuned with experimental data to reproduce experimental observations. Various phenomenological models that can output structure have been developed, differing in the number of fitting parameters[21–29]. These models work well for optimization problems where the parameter space to be explored has a few dimensions. However, a realistic problem in laser processing often involves a variety of processing parameters, as well as diverse processing methods where parameters are dynamically modulated, which have been made available by recent advances in laser processing[30–32].

Optimization across parameter-modulated conditions involves a much higher-dimensional search space than for parameter-constant conditions, since parameters can take arbitrary values within their respective range for every step in the process. An increase in dimensions leads to an exponential increase in the number of possible conditions. For example, in a simple case where a parameter value can take $M$ discrete values and the number of steps in the process is $N$, the number of parameter-constant conditions is $M \times N$, while the number of parameter-modulated conditions is $M^N$ (Fig. 1a). Therefore, the realization of a simulator applicable to high-dimensional optimization using conventional phenomenological approaches would require an infeasible amount of fitting data, and there has been no report on such high-dimensional optimization.

Here, we report on the demonstration of simulator-based optimization conducted across a much higher-dimensional space than that of the fitting data, in ultrashort pulse laser drilling of glass. The fabrication of high-aspect ratio microchannels in brittle materials such as glass is a key process in various fields, ranging from microfluidics[33,34] to IC packaging[35,36]. Ultrashort pulse laser drilling is a promising method for realizing such structures with speed and precision[37,38]. However, it remains a challenge to find the optimal drilling condition that produces a microchannel of given design, as the drilled structure depends greatly on numerous laser parameters[37,39–46], and effects such as nonlinear absorption and beam propagation further complicate ablation mechanisms compared to metallic materials[29–31]. In this work, we aimed to construct a laser drilling simulator which can be used to rapidly find an energy-efficient drilling condition for a microchannel of given structure. Energy efficiency is an important optimization objective in the field of laser processing, as the realization of sustainable and carbon neutral technologies is desired in various fields of machining. The simulator was constructed using an iterative deep learning scheme[47] which enables optimal search across a vast parameter space previously unachievable by conventional deep learning-based methods. Using this simulator, we conducted a virtual grid search for 46,656 different pulse energy-modulated drilling processes in less than 2 h. From the grid search result, we discovered a modulation condition 20% more energy-efficient than any of the experimental data on which the simulator was trained, and this optimal condition was successfully validated (See Supplementary Fig. 8 for an overview of the operations conducted in our work). This work verifies the potential of our deep learning-based approach for the realization of next-generation smart laser processing systems with predictive capabilities.

## Results

**Training and simulation scheme of the deep learning-based simulator**. In this section, we provide an overview of our deep learning-based scheme for developing a simulator for deep hole drilling of glass. Deep learning[48] is a subfield of machine learning, where a multi-layer function called the deep neural network (DNN) is used to approximate input-output relationships. Its advantage over other machine learning methods is that the DNN can be designed and trained to directly approximate relationships between high-dimensional data such as images, which can be attributed to the massive amount of fitting parameters composing the DNN. Due to this advantage, deep learning has been applied to various tasks in laser processing such as the prediction of processing results[27,28,47], feature extraction[49,50], quality evaluation[49], and fetching the used processing parameters[51].

In order to develop a simulator for deep hole drilling of glass, we employed an iterative deep learning-based scheme where a DNN is trained to predict the processed result after a small number of irradiated pulses. Once the DNN is trained, a full drilling process consisting of many irradiated pulses can then be simulated by iteratively re-inputting the predicted output of the DNN into itself. Using this scheme, we developed a DNN which takes a pulse energy value and the current microchannel structure as inputs, and predicts the resulting microchannel structure after 10 irradiated pulses. Our scheme allows the straightforward simulation of drilling processes of arbitrary pulse number, as well as practical processes where pulse parameters such as energy and polarization are dynamically modulated[30–32]. This scheme is fundamentally different from the conventional deep learning-based scheme for laser processing applications for processing result prediction[27,28] where a DNN is trained to directly predict the final outcome of the entire process without iteration.

A conceptual diagram of the training process and simulation method is shown in Fig. 1. First, the training data for the DNN was collected (Fig. 1b). Laser drilling was performed on a sheet of borosilicate glass with an ultrashort pulse laser (800 nm,

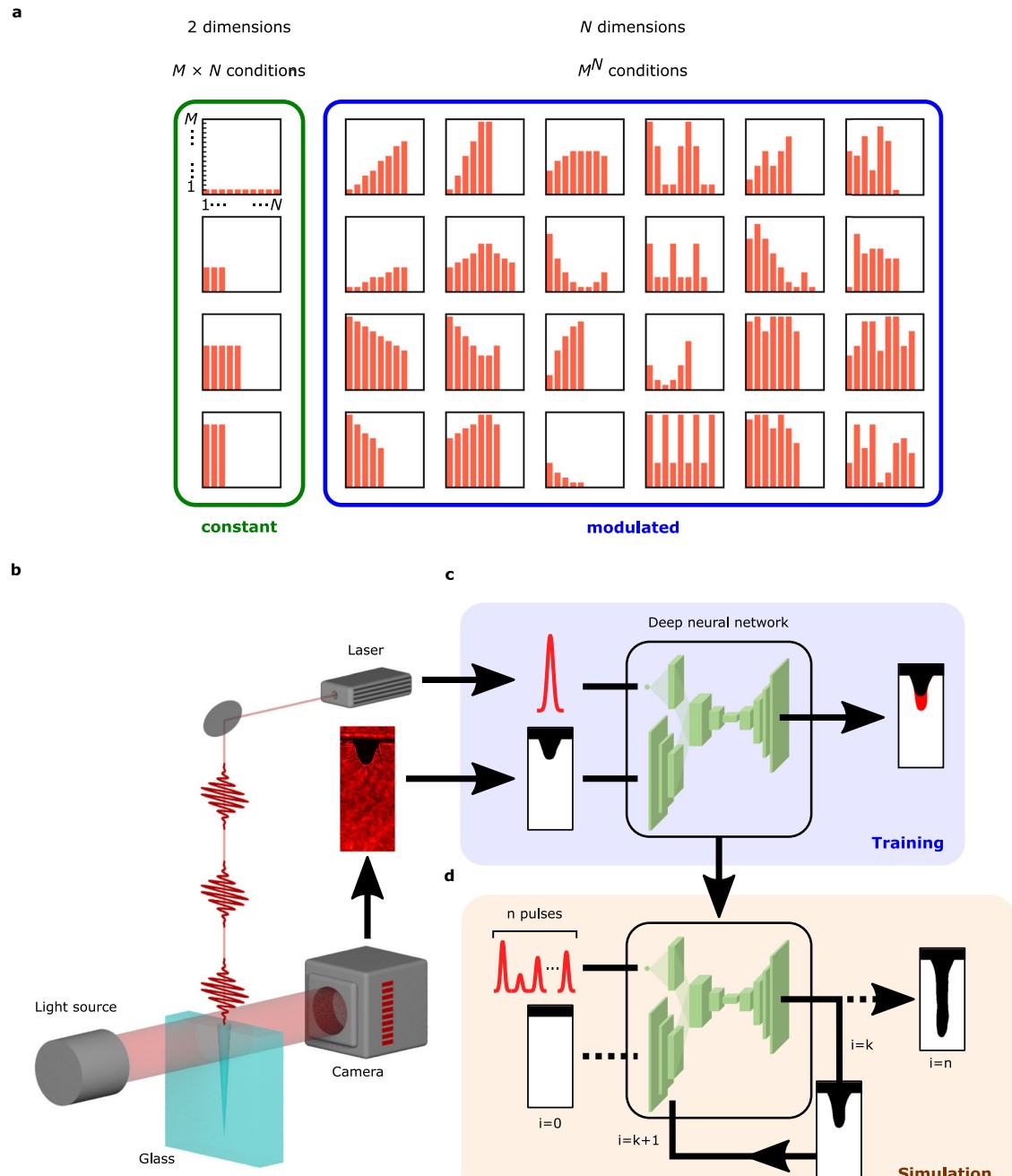

**Fig. 1 Diagram of the exponential expansion of the optimization space for higher dimensions, and the construction of a simulator for optimization in such spaces. a** Schematic comparison between parameter-constant conditions (green box) and parameter-modulated conditions (blue box). **b–d** Training and simulation scheme of the DNN simulator. **b** Collection of the training data using a high-speed camera. **c** Training process of a deep neural network (DNN). The red region in the right vacancy map indicates the newly drilled area of the microchannel. **d** Iterative simulation using the trained DNN.

35 fs, 1 kHz repetition rate), during which a series of transmissive images of the drilled microchannel was captured with a high-speed camera. The camera was synchronized with the repetition of the laser, enabling real-time acquisition of the microchannel image after every irradiated pulse (See Methods for details on experimental setup). Transmissive images were collected for drilling processes of 1000 pulses and constant pulse energy. The pulse energy was fixed to 25 different values, ranging from 10 µJ to 250 µJ in 10 µJ increments. Three drilling experiments were repeated for each pulse energy condition, and the data collected for two trials were used to train the DNN, while the remaining data was left out for validating the DNN

during training. Once the transmissive images were collected, the structure of the drilled microchannel was extracted from each image as a black-and-white image representing the region free of glass, which we will refer to as a vacancy map (Methods). The vacancy maps were re-arranged into numerous pairs of maps 10 pulses apart (Supplementary Fig. 5), and a DNN was trained to reproduce the latter map of the pair when it received an input of the former map and pulse energy value (Fig. 1c). By using the trained DNN in an iterative manner, we were able to obtain a sequence of vacancy maps representing the transient growth of the microchannel after every 10 irradiated pulses (Fig. 1d).

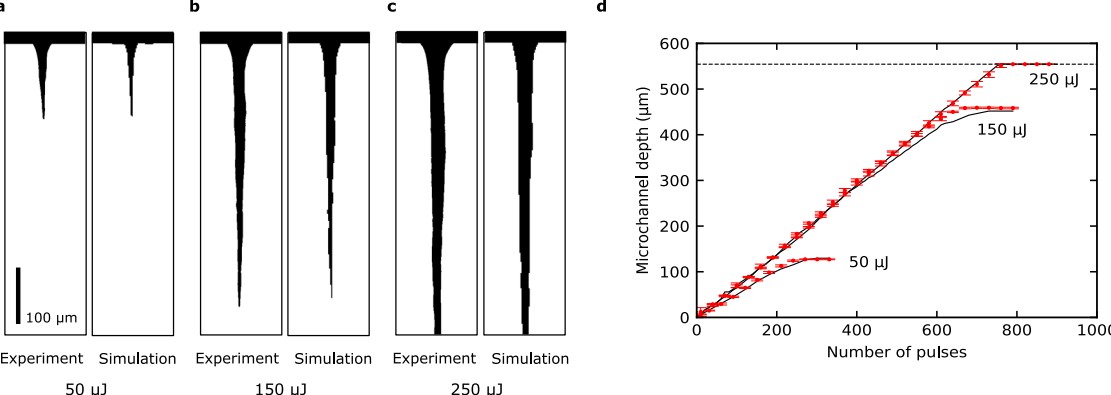

**Fig. 2 Comparison of simulation and experiment for drilling conditions included in the training data. a–c** Simulation result and experimental data of the drilled microchannel for pulse energies of **a** 50 μJ, **b** 150 μJ and **c** 250 μJ. Scale bar is 100 μm. **d** Change of microchannel depth over multiple irradiated pulses for three pulse energies. Solid lines correspond to the simulation result, and the red points correspond to the experimental data. Horizontal dashed line indicates the distance between the glass surface and bottom edge of the vacancy map, which is 554 μm. Error bars show the error of the drilled depth calculated from three experimental trials.

**Evaluating the accuracy of the DNN for trained conditions**. In order to evaluate the applicability of our DNN simulator to optimization, we first investigated whether the DNN could reproduce the drilling processes on which it was trained. While the DNN was trained to predict the change of microchannel structure over 10 pulses, the evaluation focused on whether the DNN could reproduce the final structure over several hundred pulses.

Simulations of 1000 pulses were conducted for pulse energies of 10 μJ to 250 μJ in 10 μJ increments, and the predicted vacancy maps were compared to that of the actual experimental data. The simulation results and corresponding experimental data for drilling processes at three different pulse energies (50 μJ, 150 μJ and 250 μJ) are shown in Fig. 2 and Supplementary Movies 1–3. The visual comparisons (Fig. 2a–c) show that the final microchannel shape predicted by the simulator reproduces the experimental data. The transient evolution of the microchannel depth was also compared between simulation and experiment (Fig. 2d). For pulse energies 50 μJ and 150 μJ, the simulation reproduces the linear rise and eventual saturation of the microchannel depth. As for the plot for 250 μJ, the saturation of the microchannel depth is due to the microchannel approaching the bottom edge of the vacancy map (Fig. 2c) which was limited by the imaging range of the high-speed camera (Methods); the actual saturation of depth occurred beyond 800 pulses. As a quantitative method to evaluate the accuracy of the DNN simulator, we chose to calculate the relative error of drilled depth. The average relative error for all predicted microchannels with depths between 50 μm and 554 μm (Methods) was found to be 3%, indicating the high accuracy of the DNN simulator for energy-constant drilling conditions.

**Evaluating the ability of the model to simulate pulse energy-modulated drilling processes**. Once confirmed that the DNN could reproduce the change of microchannel structure over several hundred pulses for energy-constant drilling conditions included in the training data, we evaluated the predictability of the DNN for conditions where the pulse energy is modulated during the drilling process. Here, the difference between energy-constant conditions and energy-modulated conditions should be noted in detail: although the processing parameters (pulse energy and number of pulses) and their range are the same for both types of conditions, the latter has a much higher degree of freedom, or number of dimensions, since the pulse energy value can take a different value every 10 pulses (Fig. 1a). Therefore, the evaluation

of predictability for energy-modulated conditions was a test of whether our simulator could make valid predictions across a much higher-dimensional space than that from which the training data was collected, which has not been achieved by conventional deep learning-based schemes as well as other interpolation methods[26,52,53] applied to laser processing.

Two basic energy modulations were chosen as test cases: an upward modulation where the pulse energy increases monotonically during the process, and a downward modulation with the opposite feature. The pulse energy was set to increase (or decrease) every 200 pulses, and the increase (or decrease) was repeated 4 times to produce a sequence of 800 pulses (Fig. 3a, c). Both modulations shifted between 4 discrete pulse energy values (37 μJ, 125 μJ, 213 μJ and 250 μJ). A third condition, where the pulse energy is constant (125 μJ) and the total pulse energy is equal to the other two conditions, was chosen as a reference (Fig. 3b). The energy-modulated drilling processes were simulated by setting the input pulse energy of the DNN to the prescribed value for each iteration. The experimental validation of the modulation conditions was conducted using the same experimental setup used for training data collection (see Methods).

The simulation result and corresponding experimental result for the three conditions are shown in Fig. 3d–f. Comparing the experimental results in Fig. 3d–f, it can be observed that different modulations produce microchannels of different structure even under equal total pulse energy. This can be attributed to the fact that the distribution of energy deposited in the glass depends greatly on the structure of the microchannel created by the previous pulses, since the microchannel shape alters both propagation and absorption characteristics of the subsequent pulses. Figure 3 shows that the simulator was able to reproduce the tendency that a downwards modulation produces a deeper microchannel than an upwards modulation. The simulator was able to predict the final drilled depth for the three conditions with an average relative error of 5% in depth. The simulator was not able to reproduce certain features of the drilling process such as the bending of the microchannel in Fig. 3f, which can be attributed to the re-deposition of ablated material on the inner walls of the microchannel, creating an asymmetry in the propagation of subsequent pulses[41,43].

**Optimization**. Once we confirmed that our DNN simulator could make accurate predictions for basic energy-modulated

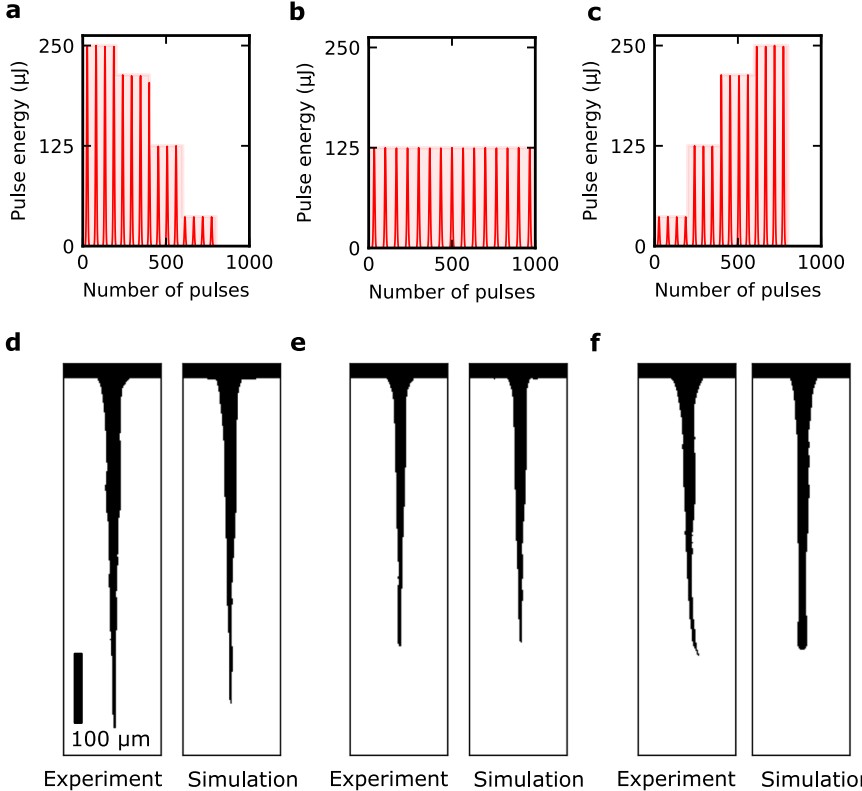

**Fig. 3 Comparison of simulation and experiment for three different energy-modulated conditions with equal total energy. a–c** Pattern of (**a**) downward modulation, (**b**) constant, and (**c**) upward modulation. **d–f** Experimental result (left) and simulation result (right) for (**d**) downward modulation, (**e**) constant, and (**f**) upward modulation. Scale bar is 100 µm.

conditions not included in the training data, we applied our simulator to an optimization problem involving a high-dimensional search space consisting of diverse modulation conditions. The objective of optimization was chosen to be the total pulse energy required to drill a microchannel of a given depth, as energy efficiency is an object of industrial interest in the field of laser processing[54]. We conducted an extensive grid search simulation across a multitude of energy-modulated conditions, and searched for the condition that drills a microchannel with the least total energy.

The grid simulation was conducted as follows. All modulations consisted of 1200 pulses and were chosen to be 6-step sequences, each step consisting of 200 pulses. The energy value of each step was taken from 6 discrete energy values for a total of $6^6 = 46,656$ modulation patterns. It is worth noting that, due to the increase in dimensions, this quantity is considerably greater than the number of energy-constant drilling conditions the DNN was trained on, which was 25. One hundred and twenty vacancy maps were obtained from each modulation simulation, corresponding to each of the 120 iterations for the DNN to simulate a 1200-pulse drilling process. The time required for the DNN to simulate all modulations was under 2 h on typical consumer-grade hardware. The data collection for 25 energy-constant drilling conditions took 2 h with our setup, including the time required for translating and replacing the glass sample; therefore, an experimental grid search covering all 46,656 drilling conditions would take 5 months. Several examples of the simulated modulation conditions and the corresponding results from the grid search are shown in Supplementary Fig. 7. The entire grid search result was then visualized by plotting the predicted microchannel depth versus total pulse energy for each simulated modulation.

The grid search visualization result is shown in Fig. 4a. Each blue point in the figure corresponds to a simulation result under a different energy modulation or number of pulses. Here, the plotted grid simulation conditions are limited to those that our experimental setup was capable of experimentally validating, the number of such conditions being 950 out of 46,656 (see Methods). Considering the 120 intermediate simulation results for each of the 950 conditions, a total of 29,560 results with unique modulation and number of pulses were obtained and plotted. As a reference, the experimental results for constant drilling conditions included in the training data (10 µJ to 250 µJ in 10 µJ increments) are plotted as a cluster of red points. The blue cluster is distributed more broadly than the red cluster, showing that energy-modulated processes produce diverse structures that cannot achieved by energy-constant drilling, according to the DNN simulator.

The plot was utilized to find the most energy efficient condition that realizes a given microchannel depth. The red circle and red triangle correspond to the most and least energy-efficient condition to realize a microchannel depth of 500 µm among all constant drilling conditions, respectively. For both conditions, the simulation results agree well with experiment, as shown in Fig. 4c, d. The most energy-efficient condition among modulated conditions for 500 µm is indicated by a blue circle, which lies left of the red circle by 20%. This prediction implies that a 500 µm-deep microchannel could be achieved with 20% better energy efficiency than constant drilling by modulating the pulse energy. The optimal condition predicted by the DNN was validated by experiment, as shown in Fig. 4b and Supplementary Movie 4. The DNN was able to predict the final structure of the microchannel with a relative error of 6% in depth. Such optimal

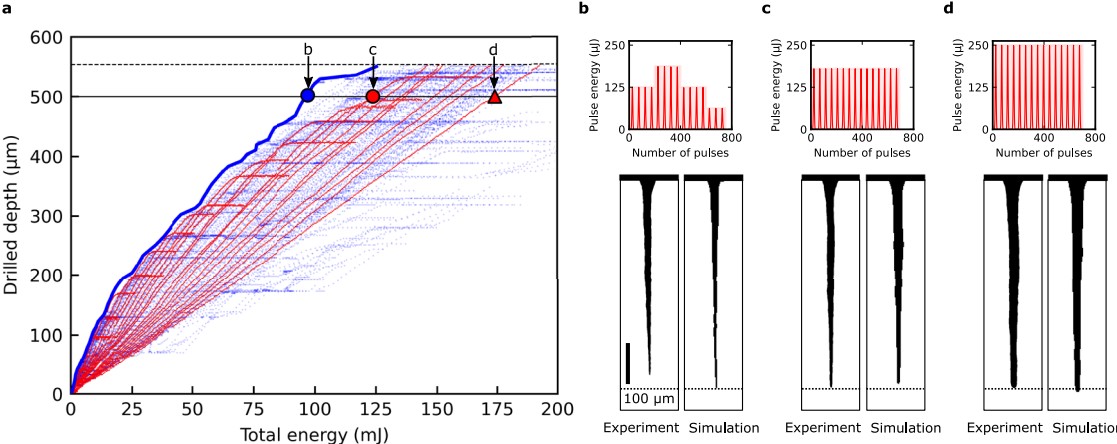

**Fig. 4 Results of the virtual grid search and optimization. a** Visualization of grid search result. Red plots correspond to the simulation results of constant drilling conditions included in the training data. Blue plots correspond to modulated conditions not included in the training data. Horizontal solid line corresponds to a depth of 500 μm. Horizontal dashed line indicates the distance between the glass surface and bottom edge of the vacancy map, which is 554 μm. The blue circle indicates the most energy-efficient condition among modulated conditions for realizing a 500 μm-deep microchannel. The red circle and red triangle indicate the most and least energy-efficient condition among constant drilling conditions, respectively. **b–d** Pulse energy sequence (top), simulation result (bottom right), and experimental validation result (bottom left) for (**b**) most efficient energy-modulated condition, (**c**) most efficient energy-constant condition, and (**d**) least efficient energy-constant condition to produce a microchannel of 500 μm depth. Dotted lines are drawn at a depth of 500 μm. Scale bar is 100 μm.

modulation conditions exist for other depths, as shown by the blue line tracing the left edge of the blue cluster.

## Discussion

We succeeded in discovering and validating an optimal modulation condition outside the training data, with an energy efficiency unachievable by any constant drilling condition. Since the depth of the microchannel was the main feature chosen for evaluating the validity of our DNN simulator, it may seem that a simulator simply predicting depth would have been sufficient for optimization; however, it is to be emphasized that the change of structure caused by subsequent pulses depends greatly on the structure of the initial microchannel, and the drilling process cannot be described simply by depth. Moreover, our simulator can easily be utilized for optimizing quantities other than depth such as aspect ratio or inner wall curvature. Although a grid search was employed in this study to demonstrate the applicability of our simulator to high-dimensional optimization, the employment of a more time-efficient algorithm is of high interest, since even a virtual grid search becomes less feasible with the expansion of the search space, due to computational limits.

The application of our simulator also extends to other important tasks in laser processing, such as rapidly investigating the effect of processing parameter fluctuations on the final processing result to determine processing windows. On a broader scale, our iterative deep learning-based scheme for deep hole drilling can be used to develop even more versatile simulators, which take multiple pulse parameters as inputs, and predicts features other than structure such as distribution of peripheral damage. This research verifies the potential of our simulator development approach for the realization of next-generation smart laser processing systems.

## Methods

**Experimental setup.** The sample used was a thin borosilicate glass sheet from Matsunami Ltd, with a planar dimension of 18 mm × 18 mm and a thickness of 0.12–0.17 mm. Laser drilling was performed using a mode-locked, regeneratively amplified Ti:Sapphire laser system (Astrella, Coherent), operating at 800 nm with 35 fs pulse duration and 1 kHz repetition rate. The pulse was focused onto the surface of the thin side of the glass using a plano-convex lens with a focal length of 150 mm. The diameter of the beam spot was measured to be 30 μm. The glass was

illuminated by a 637 nm laser diode from a direction perpendicular to that of the drilling pulse, and the transmitted light was imaged onto a high-speed camera with a sensor of 1024 pixels × 1024 pixels (FASTCAM Mini AX50, Photron). The camera was synchronized with the repetition of the laser using a delay generator (DG645, Stanford Research Systems), enabling real-time acquisition of the drilled microchannel image after every irradiated pulse. A polarization beam splitter and a half-waveplate mounted on a mechanical rotational stage was inserted before the focusing lens for pulse energy adjustment. The imaging range of the camera was 614 μm × 614 μm, and transmissive images of 144 μm width and 576 μm depth were cropped from the full range to be later converted to vacancy maps. See Supplementary Fig. 1 for a diagram of the experimental setup.

**Generation of vacancy maps.** The main objective of converting the collected transmissive images to vacancy maps was to prepare a dataset that only contained structural information on the microchannel, and therefore, prevent the DNN from making predictions based on features irrelevant to the drilled microchannel. Such irrelevant features include damage formed on the lateral surface of the glass, and interference patterns caused by back reflection between the glass and optical components (Supplementary Fig. 2). The lateral damage was formed during the drilling process but remotely from the microchannel, as was observed by a scanning electron microscope (Supplementary Fig. 3). The spatial extent of the lateral damage depended on pulse energy and were thought to be attributed to the interaction of the lateral surface with either the refracted portion of the pulse or shockwaves formed at the tip of the microchannel. Although lateral damage was removed from the vacancy maps to limit the scope of the simulator to microchannel structure, the inclusion of such features in the training data to construct a simulator applicable to damage prediction and reduction is an interesting problem to be discussed in future works.

Polarization microscopy was also conducted on a few sample microchannels, in order to image microscopic cracks extending directly from the microchannel, and to investigate any possibilities of erroneously detecting such cracks as ablated regions in the transmissive image. As a result, we found that the cracks visible in the polarization image did not appear as clearly as the ablated region in the transmissive image, therefore eliminating the possibility of mistaking a crack for an ablated region (Supplementary Fig. 9).

An additional advantage of training the DNN on vacancy maps was that the predicted outputs could be easily interpreted and analyzed. Since the region of black pixels represents the spatial extent of the drilled microchannel, basic quantities such as channel depth and width profile could be calculated by measuring the dimensions of the black region.

Vacancy map generation was conducted by inputting the transmissive images into a DNN different from the DNN simulator. This DNN took a U-Net structure[55] (Supplementary Fig. 4), and was trained in advance on 600 pairs of transmission images and manually extracted vacancy maps. The weights of the DNN were optimized using the Adam algorithm[56] to minimize the mean-squared error, which was calculated by averaging the squared error of each pixel of the vacancy map. The training was run for 300 epochs.

Once all transmissive images were converted to vacancy maps, the maps were re-arranged into numerous pairs of maps 10 pulses apart to be used as the training data for the DNN (Supplementary Fig. 5). The interval value of 10 irradiated pulses was chosen so that the difference in depth between the input and output microchannel during the early stages of the drilling process was greater than 5 μm, corresponding to a few pixels in the vacancy map; when the DNN was trained on pairs that were only 1 irradiated pulse apart, the DNN made erroneous predictions where the microchannel would either recede midway or not evolve at all, and this effect was attributed to the inability of the DNN to detect the few pixels' difference of depth between the input and output vacancy map. The training was conducted on a graphics processing unit (RTX A6000, NVIDIA).

**Calculating the relative error of predicted depth**. The relative error of depth was calculated by dividing the depth difference between the predicted and target microchannel by the depth of the target microchannel. The depth of both predicted and target microchannels were measured by counting the number of black pixels from the surface to the bottom of the microchannel in the vacancy map. As for the training accuracy of the DNN simulator, the relative error of depth was calculated for all predicted microchannels with depths between 50 μm and 554 μm, and averaged. The lower limit of 50 μm was chosen since a typical microchannel for industrial application has an aspect ratio exceeding unity, and the average width of the microchannels produced in this experiment was 50 μm. The upper limit of 554 μm corresponds to the distance between the glass surface and bottom edge of the vacancy map.

Although the main metric used in this research to evaluate the error of the simulation result was error of depth, an alternative and more 2D-oriented method is to compare the volume of the microchannel between the predicted and target image. A volume of a microchannel can be calculated by adding up the square of the microchannel halfwidth at each depth, assuming that the microchannel is axially symmetrical. Using this method, the relative error of volume can be calculated by dividing the difference in volume by the volume of the target microchannel. For example, the relative volume error for the simulation result in Fig. 4d is 3%.

**Training specifications of the DNN simulator**. The structure of the DNN simulator was chosen to be a modified version of the U-Net structure, where a multi-layer perceptron branch taking a pulse energy value as input was merged at the encoding layer of a conventional U-Net structure which took a vacancy map as input (Supplementary Fig. 6). Therefore, a single training data consisted of two inputs and one output: the initial vacancy map and the pulse energy as the two inputs, and the resulting vacancy map as the single output. In addition to the vacancy maps generated from the experimental data, zero-energy data were artificially created and added to the training dataset, where the input energy value was zero, and the input and output vacancy maps were identical. This augmentation was based on the physical assumption that the structure of a microchannel does not change when there is no incident pulse, i.e. the pulse energy is zero. The rapid and efficient preparation of training data is a key ingredient for an effective deep learning application, and data augmentation methods such as the one conducted in our work are powerful techniques for achieving this, especially when there is a limit to how rapidly one can collect data by experimental means. Once augmented, the full dataset consisted of 28,167 data. The DNN was trained on this dataset for 300 epochs. The weights of the DNN were optimized using the Adam algorithm to minimize the mean-squared error between the predicted map and target map.

**Specifications of the experimental validation of the energy-modulated processes**. For the experimental validation of energy-modulated drilling processes, an additional polarization beam splitter and half-wave plate were inserted into the setup (Supplementary Fig. 1), and the waveplate was mounted on an electrical rotational stage (ELL14K, Thorlabs). The rotation angle of the stage per single step was fixed. The rotation direction for each step was controlled via a microcomputer board (Arduino Uno Rev 3). The actual pulse modulation was measured by monitoring the residual transmitted light from a turning mirror with an oscilloscope (DSOX3054T, Keysight). It is to be noted that the modulated drilling processes realized with this modulation system were not included in the training data, and only used for the validation of the DNN simulator.

**Specifications of the grid search simulation**. The energy values of each modulation step were taken from 6 discrete energy values: 17 μJ, 62 μJ, 125 μJ, 188 μJ, 233 μJ and 250 μJ. These values correspond to the pulse energy values that could be obtained by rotating the waveplate from 0° to 45° in $45°/6 = 7.5°$ increments, with the maximum pulse energy set to 250 μJ. The number of pulses per modulation step was set to 200, corresponding to a duration of 200 ms. The value of 200 ms was chosen to be sufficiently greater than the time required for the waveplate to rotate 7.5°, which was measured to be 10 ms. The 950 simulated conditions plotted in Fig. 4 were limited to conditions where the pulse energy values of each step were adjacent to or equal to one another, since the waveplate could only be rotated in three ways (clockwise, anti-clockwise, or static) for each step. $950 \times 120 = 114,000$ simulation results were obtained from the grid simulation including the intermediate vacancy maps, and duplicate conditions with the same number of pulses and energy modulation were excluded to yield 29,650 unique simulation results. For most energy modulations with multiple energy values close to 250 μJ (Supplementary Fig. 7d, i), the microchannels reached the bottom edge of the map at around 800 pulses, and after 1200 pulses the microchannel exceeded the bottom edge. Therefore, the majority of the microchannels shallower than the bottom edge was drilled in less than 800 pulses, which was also the case for the modulation conditions in Fig. 4b–d.

**Computer specifications**. The training of the DNN models and the grid search optimization was conducted on a computer with an AMD X399 motherboard, whose power was supplied by a 1500 W power unit (AX1500i, Corsair). The motherboard hosted a CPU with 64 threads and a base clock frequency of 3 GHz (Ryzen Threadripper 2990WX, AMD), eight 16 GB memories (CD16G-D4UE2666, Century Micro), and two GPUs (RTX A6000, NVIDIA). Only one of the two GPUs was used in this work.

## Data availability
The source data for Fig. 2d and Fig. 4a is provided as Supplementary Data 1 and Supplementary Data 2, respectively. Other data used in this paper are available from the corresponding author upon reasonable request.

## Code availability
The DNN models were written in Python and are available from the corresponding author upon reasonable request.

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

## Acknowledgements
This work was supported in part by New Energy and Industrial Technology Development Organization (TACMI project, P16011); Council for Science, Technology and Innovation; Cabinet Office, Government of Japan; National Institutes for Quantum and Radiological Science and Technology (Cross-ministerial Strategic Innovation Promotion Program); Ministry of Education, Culture, Sports, Science and Technology (JPMXS0118067246 and JPMJCE1313).

## Author contributions
K.S, S.T., H.S and Y.K. conceived the study. K.S. and S.T. performed the experiments and analyzed the data. K.S, S.T., H.S and Y.K. wrote the paper.

## Competing interests
The authors declare no competing interests.
