## [Peer review file · Communications Engineering]

Reviewers' comments:

Reviewer #1 (Remarks to the Author):

The work presents using deep learning to simulate laser pulse drilling and do so with varying laser power input and improved efficiency. The results are impressive, and the work is an important addition to the field, especially the novel use of perceptrons. I recommend the work for publication so long as the following comments are addressed:

1. Whilst the authors reference an article about the need for efficient machining, to allow a reader to put into context the 20% efficiency, it would be useful to the reader to include a sentence or two in the introduction about what potential savings and speedup correspond to in certain processes or industry.
2. Also for the introduction, it would be useful to the reader if the following seminal work in the laser machining and deep learning field was referenced:
 - a. Mills, Ben, et al. "Image-based monitoring of femtosecond laser machining via a neural network." *Journal of Physics: Photonics* 1.1 (2018): 015008.
3. Since the shape of the generated image is important as well as the depth, it would be useful for the reader if an error between the actual and simulated images could be produced?
4. When a single perceptron taking a pulse energy value is connected to 25,600 perceptrons, does this mean the 25,600 array has the same value in each array element?
5. As I understand it, the non-peripheral damages were removed from the vacancy maps. Out of interest to perhaps be discussed in future work, would it be worth keeping the non-peripheral damages in the vacancy map to allow a neural network to be able to avoid or reduce damage in further pulses?
6. It would be useful for anyone wishing to implement their own deep learning to have an idea of the computer specifications (CPU, RAM and power supply etc.) that was used to host the A6000 GPU.
7. In Fig. 7, the images and text of the figure appear to be blurry. Please increase correct this for ease of viewing.

Reviewer #2 (Remarks to the Author):

This is an interesting work in the field of laser drilling and parameters optimization. The authors propose a DNN model for the prediction and optimization of the laser drilling process. Also, they designed a practical method to generate the experimental data for training and prediction, which validates the effectiveness of their model. Their model shows a good predictive ability and adaptation in a wide range of laser processing parameters. However, some minor problems need to be clarified.

1. The authors state that they are the first demonstration of simulator-based optimization in ultrashort pulse laser drilling of glass. They should check the following reference:

Liao, K., Wang, W., Mei, X. et al. Shape regulation of tapered microchannels in silica glass ablated by femtosecond laser with theoretical modeling and machine learning. *J Intell Manuf* (2022).
<https://doi.org/10.1007/s10845-022-01950-z>

2. In the Supplementary Fig. 2, the authors present their image processing method. However, for laser drilling of glass, sometimes the material modification or the crack of the glass looks like the ablated region of the drilling hole's bottom, which means the hole is not really getting though even the image looks like it has already reached the depth. The authors should clarify the potential error and get the real depth of the laser drilling hole.
3. In discussion section, it is exaggerating to state that the authors model conducted across a parameter range exceeding the fitting data. However, the model is still under the DNN framework and

it is constrained in the range of pulse energy. The authors just have some novelty implementation of their model.

Reviewer #3 (Remarks to the Author):

The major claim of the paper is the first demonstration of simulator-based optimization in ultrashort pulse laser drilling of glass, applying a deep learning (DL) approach. It is also claimed that, through a rapid but significant parameter search, the simulator discovered a modulation condition 20% more energy-efficient than any from experimental data supplied for simulator training. The optimal condition found is said to be successfully validated, thus verifying the predictive capability potential of the deep learning-based approach for future smart laser processing systems. Another aspect of novelty claimed is the iterative method used here, whereas previous (DNN) models had been only predictive.

The claims appear to be novel, though it would seem more correct to assign the claims to the specific material (glass) and set of conditions explored; whereas the current paper title and some conclusions drawn imply a more general claim. From literature reviewed, the following aspects would appear to potentially compromise the novelty and/or the generality of what is claimed. Firstly, a critical result already identified for ultra-short pulse laser (USPL) drilling / machining has been real-time process feedback achieved by the capture and integration of camera images during the process [Mills B, et al (2019), J. Phys. Photonics, 1, 015008]. The authors already reported on the use of interferometric surface speckle patterns as input to a convolutional neural network (CNN) [Tani S, et al (2020), Neural-network-assisted in situ processing monitoring by speckle pattern observation, Optics Express 28(18), 26180-26188]. Secondly, the authors recently published some papers on the same theme. These include a paper in Nature Scientific Reports with a similar title [Tani, et al; Ref. #44]. On review, that appears to be a broader study, yet the approach seems to be the same and the materials studied included, amongst others, sapphire and GorillaGlass3. Also, [Bamoto K, et al (2022), Autonomous parameter optimization for femtosecond laser micro-drilling, Optics Express 30(1), 243-254] and Kobayashi, et al (2021) in IEEE [Ref. #4]. The authors should make it clear to the reader how this present submitted manuscript and its claims are distinguished from these other published works and their findings. There looks to be plenty of distinction in results, presentation, analysis etc. between these papers, but not in the DL/DNN methods used. A paper on deep learning applied to the laser welding process has been published [Gunther J, et al (2016) Mechatronics 34, 1-11]; the authors should include this in their review of literature and, if appropriate, qualify how their work builds on advances already made in applying DL to the broader discipline of laser processing.

The paper should be of strong interest to others in the field. The development of deep learning models/simulators is a good fit for manufacturing, since processes generate large amounts of data from machine sensors, yet currently most firms do not utilise this data effectively. Leading analysts forecast a trillion-plus USD market for deep learning models in both manufacturing and its wider sphere of supply chain management and logistics. There is increasing demand for such cyber-physical systems to enable new capability for: predictive maintenance to prevent in-production failures; predictive analytics for process optimisation; eliminating bottlenecks in the production cycle (e.g. materials supply); product development; quality assurance; and robotics, to handle complicated and hazardous processes / materials.

Will the paper influence thinking in the field?

The model/simulator studied only considers 2 parameters, yet the laser drilling process using USPLs is known to be dependent on several applied parameters and conditions. Hence, a concern is the true novelty of what is achieved so far, and (realistically) how the approach could be viably translated to multiple parameters, more representative of a USPL drilling application. That said, the achievements being reported are certainly an important step in this direction. The authors state that the vast

majority of previous models have not been validated outside the range of their experimental tuning parameters, and thus, their applicability for optimization is questionable. This could also be said of the case being reported here.

A challenge for deep learning in such a process is the large amount of data needed for accurate training of the model/simulator, with each item of data 'handled' requiring a process parameter to be changed (thus modifying the laser-material interaction) and an associated measurement to quantify the result and its level of success against chosen criteria. Collecting such data could take weeks; hence, not feasible. 'Data augmentation' methods are needed e.g. of captured images. For the claims to be convincing, there should at least be appropriate discussion to show an awareness of this challenge.

The authors claim that the simulator discovered a modulation condition 20% more energy-efficient than any from experimental data supplied for simulator training. From knowledge of laser drilling in both non-thermal (USPL) and thermal regimes, it seems intuitive and actually not surprising that a temporal variation in pulse energy of the form shown in video #4 gives the most effective result (increasing from an initial energy value, then falling away as a 'tail'). However, from my understanding, it is not clear if the simulator was trained to seek out such a form. What was the decision-making applied at the start and during the process?

How does the model respond to anomalies or an erroneous change in one laser parameter e.g. applied fluence? What is its sensitivity to laser parameter fluctuations? This would seem important for future application to high precision ultrashort pulse machining involving highly nonlinear interaction mechanisms under industrial / engineering settings and conditions.

I cannot offer an opinion on whether there other experiments that would strengthen the paper further, how much would they improve it, and how difficult are they likely to be. However, addressing the points already made should help negate any need for further experimental work. There already appear to be ample results worthy of publication.

Overall, the claims appear to be appropriately discussed in the context of previous literature. My opinion is that the manuscript is acceptable in its present form, taking into consideration the above comments and addressing concerns raised therein.

The manuscript is clearly written and appears to follow the required format. However, the current title is considered too broad to align with the claims of its findings. For example, a more appropriate title might be: 'A deep learning-based predictive simulator for 2-parameter optimization of ultrashort pulse laser drilling'.

Regarding whether the manuscript could be shortened to aid communication of the most important findings, it is not clear to me how this could be achieved without removing important detail.

With some clarification to address the points raised, the authors' claims made can be justified and the article would not be 'overselling' these.

Overall, the authors appear to have been fair in their treatment of previous literature and this looks well-balanced for a full research article (compared to a review article). Some readers may benefit from including a little more insight into how the DL approach fits within – but is distinct from – the wider 'set' of machine learning (ML) and artificial intelligence (AI). An example could be the set diagram used by previous authors [see Mills, B., Grant-Jacob, J.A.: Lasers that learn: the interface of laser machining and machine learning. IET Optoelectron. 15(5), 207–224 (2021). <https://doi.org/10.1049/ote2.12039>]. Setting the work and its findings in the wider context of intelligent laser processing, there have been some relevant studies reported in additive manufacturing (specifically the selective laser melting or laser powder bed fusion process) which could

be reviewed and mentioned if/as appropriate [Jayasinghe S, et al (2020), 'Automatic quality assessments of laser powder bed fusion builds from photodiode sensor measurements', *Progress in Additive Manufacturing* (2022) 7:143–160 <https://doi.org/10.1007/s40964-021-00219-w>]

Have the authors provided sufficient methodological detail that the experiments could be reproduced? See comment below.

The statistical analysis of the data appears to be sound.

Should the authors be asked to provide further data or methodological information to help others replicate their work? (Such data might include source code for modelling studies, detailed protocols or mathematical derivations).

Detail of code would not be of interest / relevance here, but the operations carried out on the data should be clear. Fig. 1 already seeks to explain this. Perhaps a decision tree / flow diagram that could convey this in general terms for the journal readers, plus some brief detail of the numerical operations. It seems that more of this kind of detail was included in the authors' other recent publications on the topic.

Reply to reviewer's comments for Manuscript COMMS-22-0133-T "A deep learning-based predictive laser processing simulator for parameter optimization"

We would like to thank the editor and referees for the time and effort dedicated to providing valuable feedback on our manuscript. In our revision, we took into account all specific and technical comments by the reviewers as detailed below. In this reply, *italicized texts* correspond to the reviewer's comments, **blue texts** respond to the authors' replies, and **red texts** correspond to the revised/added texts in the manuscript. In the revised manuscript, the revised/added texts are shown in **red**, and the added figures are enclosed in **red boxes**.

Reviewer #1

The work presents using deep learning to simulate laser pulse drilling and do so with varying laser power input and improved efficiency. The results are impressive, and the work is an important addition to the field, especially the novel use of perceptrons. I recommend the work for publication so long as the following comments are addressed:

Comment #1

Whilst the authors reference an article about the need for efficient machining, to allow a reader to put into context the 20% efficiency, it would be useful to the reader to include a sentence or two in the introduction about what potential savings and speedup correspond to in certain processes or industry.

Reply

Thank you for the suggestion. In accordance with your suggestion, we have added a sentence discussing the importance of energy-efficiency in various fields of machining including laser processing for the realization of a sustainable and carbon-neutral society:

Changes

Lines #74-76:

Energy efficiency is an important optimization objective in the field of laser processing, as the realization of sustainable and carbon neutral technologies is desired in various fields of machining.

Comment #2

Also for the introduction, it would be useful to the reader if the following seminal work in the laser machining and deep learning field was referenced:

a. Mills, Ben, et al. "Image-based monitoring of femtosecond laser machining via a neural network." Journal of Physics: Photonics 1.1 (2018): 015008.

Reply

Thank you for suggesting this reference. We have added this work to the references, as an exemplary work in the field of deep learning application to laser processing. The reference number is #51, and has been referenced as below:

Changes

The reference was added to the highlighted position:

Lines #97-99

Due to this advantage, deep learning has been applied to various tasks in laser processing such as the prediction of processing results^{27,28,47}, feature extraction^{49,50}, quality evaluation⁴⁹, and fetching the used processing parameters⁵¹.

Comment #3

Since the shape of the generated image is important as well as the depth, it would be useful for the reader if an error between the actual and simulated images could be produced?

Reply

Thank you for pointing this out. In accordance with your suggestion, we have calculated the pixel-by-pixel error between the images, defined here as the number of erroneous pixels divided by the “area” of the microchannel. For example, the pixel-by-pixel error of the predicted microchannel in Figure 4d is 20%. However, we would like to note that this method of evaluation does not necessarily represent the structure similarity of the simulated and target microchannel, as this error is highly sensitive to minor differences in the axis of the microchannel. An alternative method more robust to such differences is to compare the “volume” of the microchannels, which can be calculated by adding up the square of the microchannel halfwidth for each depth, assuming that the microchannel is axially symmetrical. This yields a volume error of 3% for the predicted microchannel in Figure 4d. We would also like to add that this error is an accumulation of smaller errors produced during the simulation, as the simulation is conducted by repeatedly inputting the predicted microchannel image into the DNN. We have added a comment on volume comparison as an alternative way to evaluate the error between the images.

Changes

Lines #358-365

Although the main metric used in this research to evaluate the error of the simulation result was error of depth, an alternative and more 2D-oriented method is to compare the “volume” of the microchannel between the predicted and target image. A volume of a microchannel can be calculated by adding up the square of the microchannel halfwidth at each depth, assuming that the microchannel is axially symmetrical. Using this method, the relative error of volume can be calculated by dividing the difference in volume by the volume of the target microchannel. For example, the relative volume error for the simulation result in Fig. 4d is 3%.

Comment #4

When a single perceptron taking a pulse energy value is connected to 25,600 perceptrons, does this mean the 25,600 array has the same value in each array element?

Reply

We apologize for the ambiguity of information on the neural network. The single input perceptron and 25,600 perceptrons are connected by a fully connected layer consisting of 25,600 trainable weights; therefore, the output value of the 25,600 perceptrons will vary in the trained neural network. The 25,600 output values are then reshaped into an 80x20x16 array to be concatenated with the intermediate output of the U-Net structure. To resolve this

ambiguity, we have added the underlined phrase below to the captions in Supplementary Figure 6:

Change #1

Supplementary Fig. 6 Caption:

...a single perceptron taking a pulse energy value is connected to 25,600 perceptrons via a fully connected layer with trainable weights...

Change #2

We have added words to the diagram in Supplementary Figure 6 so that the reader can better understand the operation conducted in the pulse energy input branch.

Comment #5

As I understand it, the non-peripheral damages were removed from the vacancy maps. Out of interest to perhaps be discussed in future work, would it be worth keeping the non-peripheral damages in the vacancy map to allow a neural network to be able to avoid or reduce damage in further pulses?

Reply

Thank you for the insightful comment. It is precisely our vision to construct a simulator which can simultaneously predict various features of the microchannel including damage distribution. The work presented here marks the first step towards this goal. We have added a sentence covering this point in the "Generation of vacancy maps" section in the Methods.

Changes

Lines #311-314:

Although lateral damage was removed from the vacancy maps to limit the scope of the simulator to microchannel structure, the inclusion of such features in the training data to construct a simulator applicable to damage prediction and reduction is an interesting problem to be discussed in future works.

Comment #6

It would be useful for anyone wishing to implement their own deep learning to have an idea of the computer specifications (CPU, RAM and power supply etc.) that was used to host the A6000 GPU.

Reply

Thank you for pointing this out. We have added a section titled "Computer specifications" to the Methods section to address this:

Changes

Lines #416-422

The training of the DNN models and the grid search optimization was conducted on a computer with an AMD X399 motherboard, whose power was supplied by a 1500 W power unit (AX1500i, Corsair). The motherboard hosted a CPU with 64 threads and a base clock frequency of 3 GHz (Ryzen Threadripper 2990WX, AMD), eight 16 GB

memories (CD16G-D4UE2666, Century Micro), and two GPUs (RTX A6000, NVIDIA). Only one of the two GPUs was used in this work.

Comment #7:

In Fig. 7, the images and text of the figure appear to be blurry. Please increase correct this for ease of viewing.

Reply:

Thank you for pointing this out. We have enhanced the resolution of Figure 7.

Reviewer #2

Comment #1

This is an interesting work in the field of laser drilling and parameters optimization. The authors propose a DNN model for the prediction and optimization of the laser drilling process. Also, they designed a practical method to generate the experimental data for training and prediction, which validates the effectiveness of their model. Their model shows a good predictive ability and adaptation in a wide range of laser processing parameters. However, some minor problems need to be clarified.

The authors state that they are the first demonstration of simulator-based optimization in ultrashort pulse laser drilling of glass. They should check the following reference:

Liao, K., Wang, W., Mei, X. et al. Shape regulation of tapered microchannels in silica glass ablated by femtosecond laser with theoretical modeling and machine learning. J Intell Manuf (2022). <https://doi.org/10.1007/s10845-022-01950-z>

Reply

Thank you for pointing out this reference. We apologize for explaining the novelty of our claim in an insufficient and misleading manner. We would like to correct our claim by stating that our work demonstrates the first simulator-based optimization conducted across a much higher-dimensional space than that of the fitting data in ultrashort pulse laser drilling of glass, which differentiates our work from Liao et al's. We would like to elaborate on this corrected claim in the following paragraphs.

An optimization problem can be characterized by two quantities: the *number of parameters* involved in the problem, and the *dimension* (or degrees of freedom) of the search space. An energy-constant drilling condition can be described by two parameters: the pulse energy and the number of pulses. Therefore, if the range of conditions explored in the optimization were limited to energy-constant conditions, the pulse energy and number of pulses would be the only two degrees of freedom, and thus the optimization problem would be 2-dimensional (See Fig. 1a which has been newly added to the manuscript). However, the actual optimization in our work was conducted across energy-modulated conditions, which means that the energy of the pulse is no longer fixed throughout the drilling process. If we allow the pulse energy to be modulated up to N times (or modulation steps), the pulse train has N degrees of freedom, and the optimization problem becomes N -dimensional.

Our claim is that we were able to conduct an optimal search across an N -dimensional space, just by training a

model on conditions belonging to a 2-dimensional space. This means that the conditions explored in the optimization (energy-modulated) is qualitatively different from those in the training data (energy-constant). This is an important distinction from Liao et al's work, where a model was trained to predict microchannel features from 4 parameters, and an optimal search was conducted in the same 4-dimensional space. An increase in dimensions also indicates an exponential increase in the number of drilling conditions in the search space: if we limit the possible energy values a modulation step can take to M discrete values, the number of energy-constant conditions is $M \times N$, while the number of energy-modulated conditions is M^N (See Supplementary Fig. 8). Overall, there is a considerable gap between the training conditions and searched conditions in our work in both quality and quantity. Therefore, we believe that the leap from 2 to N dimensions makes the extensive simulator-based optimization demonstrated in our work the first of its kind in the field of ultrashort pulse laser drilling of glass.

In order to better convey the novelty of our work, we have made the modifications below.

Change #1

We have changed the title of this paper so that it appropriately embodies our claim and no broader:

(Before)

'A deep learning-based predictive laser processing simulator for parameter optimization'

(After)

'A deep learning-based predictive simulator for high-dimensional optimization of ultrashort pulse laser drilling'

Change #2

We have added the sentence below in the introduction section, in order to address the limit of current simulation approaches more elaborately:

Lines #47-51

These models work well for optimization problems where the parameter space to be explored has a few dimensions. However, a realistic problem in laser processing often involves a variety of processing parameters, as well as diverse processing methods where parameters are dynamically modulated, which have been made available by recent advances in laser processing³⁰⁻³².

Change #3

We have inserted a new paragraph to the introduction section, covering the difference between parameter-constant drilling conditions and parameter-modulated drilling conditions, as we have explained in our reply to Comment #1:

Lines #53-62

Optimization across parameter-modulated conditions involves a much higher-dimensional search space than for parameter-constant conditions, since parameters can take arbitrary values within their respective range for every step in the process. An increase in dimensions leads to an exponential increase in the number of possible conditions. For example, in a simple case where a parameter value can take M discrete values and the number

of steps in the process is N , the number of parameter-constant conditions is $M \times N$, while the number of parameter-modulated conditions is M^N (Fig. 1a). Therefore, the realization of a simulator applicable to high-dimensional optimization using conventional phenomenological approaches would require an infeasible amount of fitting data, and there has been no report on such high-dimensional optimization.

Change #4

We have modified Figure 1 so that it contains a new sub-figure visualizing the difference between parameter-constant and parameter-modulated conditions (Fig. 1a), and also the original figure that explains the concept of our simulation scheme (Fig. 1b-d).

Change #5

We have added the suggested reference to the references. The reference number is #29. It has been referenced in the highlighted position below:

Lines #46-47

Various phenomenological models that can output structure have been developed, differing in the number of fitting parameters²¹⁻²⁹

Change #6:

We have added the phrase below:

Lines #64-65:

Here, we report on the first demonstration of simulator-based optimization conducted across a much higher-dimensional space than that of the fitting data

Comment #2

In the Supplementary Fig. 2, the authors present their image processing method. However, for laser drilling of glass, sometimes the material modification or the crack of the glass looks like the ablated region of the drilling hole's bottom, which means the hole is not really getting though even the image looks like it has already reached the depth. The authors should clarify the potential error and get the real depth of the laser drilling hole.

Reply:

Thank you for the comment. Due to the difficulties you have pointed out, we were not fully confident whether we could distinguish the ablated region from other features at first. Therefore, in order to investigate any possibilities of erroneous detection, we conducted polarization microscopy, which is a method for imaging modifications such as microscopic cracks, on a few sample microchannels. As a result, we found that cracks visible in the polarization image do not appear as clearly as the ablated region in the transmissive image, and therefore the possibility of mistaking a crack for an ablated region was minimal (See newly added Supplementary Figure 9). As we only used borosilicate glass in our work, this tendency may not necessarily hold for other types of glass.

We have made the modifications below to address this topic.

Change #1

We have added a paragraph covering this topic to the “Generation of vacancy maps” section in the Methods section.

Lines #316-321

Polarization microscopy was also conducted on a few sample microchannels, in order to image microscopic cracks extending directly from the microchannel, and to investigate any possibilities of erroneously detecting such cracks as ablated regions in the transmissive image. As a result, we found that the cracks visible in the polarization image did not appear as clearly as the ablated region in the transmissive image, therefore eliminating the possibility of mistaking a crack for an ablated region (Supplementary Fig. 9).

Change #2

We have added Supplementary Figure 9, which compares the polarization image and transmissive image of a typical microchannel, showing that the ablated regions and microscopic cracks can be distinguished in the transmissive image.

Comment #3

In discussion section, it is exaggerating to state that the authors model conducted across a parameter range exceeding the fitting data. However, the model is still under the DNN framework and it is constrained in the range of pulse energy. The authors just have some novelty implementation of their model.

Reply

Thank you for pointing out a misleading expression. We agree that the expression “exceeding the parameter range of the fitting data” was misleading, and have fixed the manuscript. Our mistake in the previous manuscript was to use the term “parameter” for both the processing parameter and the dimension of the optimization space, which should be treated as separate quantities as explained in our reply to Comment #1. Although the range of the parameter (pulse energy) does not differ between the trained conditions and the conditions explored in the optimization, the latter conditions have much higher degrees of freedom, and therefore belong to a higher-dimensional space. We have achieved a high-dimensional optimization across diverse energy-modulated conditions using our simulator that has not been demonstrated by previous works; therefore, the novelty is not limited to the implementation of the model, but also lies in the optimization demonstration as well.

In order to prevent any further misunderstandings, we have made the modifications below.

Change #1

We have added the sentences below noting the difference between energy-constant conditions and energy-modulated conditions as explained in our reply to Comment #2:

Lines #165-173

Here, the difference between energy-constant conditions and energy-modulated conditions should be noted in detail: although the processing parameters (pulse energy and number of pulses) and their range are the same for both types of conditions, the latter has a much higher degree of freedom, or number of dimensions, since the pulse energy value can take a different value every 10 pulses (Fig. 1a). Therefore, the evaluation of predictability for

energy-modulated conditions was a test of whether our simulator could make valid predictions across a much higher-dimensional space than that from which the training data was collected, which has not been achieved by conventional deep learning-based schemes as well as other interpolation methods applied to laser processing in dielectrics.

Change #2

We have modified the expression below:

Lines #258:

(Before)

“across a parameter range exceeding the fitting data”

↓

(After)

“across a much higher-dimensional space than that of the fitting data”.

Reviewer #3

Comment #1

The major claim of the paper is the first demonstration of simulator-based optimization in ultrashort pulse laser drilling of glass, applying a deep learning (DL) approach. It is also claimed that, through a rapid but significant parameter search, the simulator discovered a modulation condition 20% more energy-efficient than any from experimental data supplied for simulator training. The optimal condition found is said to be successfully validated, thus verifying the predictive capability potential of the deep learning-based approach for future smart laser processing systems. Another aspect of novelty claimed is the iterative method used here, whereas previous (DNN) models had been only predictive.

Reply

Thank you for the comment.

Comment #2

The claims appear to be novel, though it would seem more correct to assign the claims to the specific material (glass) and set of conditions explored; whereas the current paper title and some conclusions drawn imply a more general claim. From literature reviewed, the following aspects would appear to potentially compromise the novelty and/or the generality of what is claimed.

Thank you for the comment. We agree that the absence of certain references may have clouded our claim in this manuscript. We would like to elaborate on the main claim of our work by addressing the differences between this work and the references pointed out, in the following replies.

Comment #3

Firstly, a critical result already identified for ultra-short pulse laser (USPL) drilling / machining has been real-time process feedback achieved by the capture and integration of camera images during the process [Mills B, et al (2019), J. Phys. Photonics, 1, 015008]. The authors already reported on the use of interferometric surface speckle patterns as input to a convolutional neural network (CNN) [Tani S, et al (2020), Neural-network-assisted in situ processing monitoring by speckle pattern observation, Optics Express 28(18), 26180-26188].

Reply:

We agree that the collection of images using a camera and the usage of those images as training data is similar with the method we chose in our work; however, our DL application of creating a simulator for optimization differs with those of these papers. The DL applications of the two papers are summarized below:

Mills B, et al (2019), J. Phys. Photonics, 1, 015008 [newly added as Ref #51]

Construction of a DNN that returns the used parameters when given an image of the surface structure.

Tani S, et al (2020), Optics Express 28(18), 26180-26188 [newly added as Ref #50]

Construction of a DNN that extracts features such as ablated depth when given a speckle pattern of the ablated spot.

In order to give context of the different kind of DL applications to laser processing to the reader, we have referenced the above works and mentioned them as below.

Change

The references were added to the highlighted positions

Lines #97-99:

Due to this advantage, deep learning has been applied to various tasks in laser processing such as the prediction of processing results^{27,28,47}, feature extraction^{49,50}, quality evaluation⁴⁹, and fetching the used processing parameters⁵¹.

Comment #4

Secondly, the authors recently published some papers on the same theme. These include a paper in Nature Scientific Reports with a similar title [Tani, et al; Ref. #44]. On review, that appears to be a broader study, yet the approach seems to be the same and the materials studied included, amongst others, sapphire and GorillaGlass3.

Reply

We agree that the DL method used in [Tani, S et al. *Sci Rep* **12**, 5837 (2022)] is of a similar nature to what we used in this work. However, the main distinction of our work is that we constructed a simulator which can be used to solve optimization problems across *arbitrary* pulse energy modulations just by training the model on energy-constant drilling conditions. The significance of this leap from energy-constant conditions to arbitrary energy modulations is explained below.

Optimization across arbitrary energy modulations involves a much higher-dimensional search space than for energy-constant conditions, since a pulse (or block of pulses) can take an arbitrary energy value within the given range for every step in the process. An increase in dimensions leads to an exponential increase in the number of possible conditions. For example, in a simple case where a parameter value can take M discrete values and the number of steps in the process is N , the number of parameter-constant conditions is $M \times N$, while the number of parameter-modulated conditions is M^N (See newly added Fig. 1a). Therefore, the realization of a simulator applicable to high-dimensional optimization using conventional phenomenological approaches would require an infeasible amount of fitting data, and such high-dimensional optimization has not been reported in previous works including Tani, S et al.

To clearly address the novelty of the optimization conducted in our work, we have made the modifications below.

Change #1

We have added the sentence below to the introduction, to address the limit of current simulation approaches more elaborately:

Lines #47-51

These models work well for optimization problems where the parameter space to be explored has a few dimensions. However, a realistic problem in laser processing often involves a variety of processing parameters, as well as diverse processing methods where parameters are dynamically modulated, which have been made available by recent advances in laser processing³⁰⁻³².

Change #2

We have inserted a new paragraph to the introduction, covering the difference between parameter-constant drilling conditions and parameter-modulated drilling conditions.

Lines #53-62

Optimization across parameter-modulated conditions involves a much higher-dimensional search space than for parameter-constant conditions, since parameters can take arbitrary values within their respective range for every step in the process. An increase in dimensions leads to an exponential increase in the number of possible conditions. For example, in a simple case where a parameter value can take M discrete values and the number of steps in the process is N , the number of parameter-constant conditions is $M \times N$, while the number of parameter-modulated conditions is M^N (Fig. 1a). Therefore, the realization of a simulator applicable to high-dimensional optimization using conventional phenomenological approaches would require an infeasible amount of fitting data, and there has been no report on such high-dimensional optimization.

Change #3

We have modified Figure 1 so that it contains a new sub-figure visualizing the difference between parameter-constant and parameter-modulated conditions (Fig. 1a).

Comment #5

Also, [Bamoto K, et al (2022), Autonomous parameter optimization for femtosecond laser micro-drilling, Optics Express 30(1), 243-254] and Kobayashi, et al (2021) in IEEE [Ref. #4]. The authors should make it clear to the reader how this present submitted manuscript and its claims are distinguished from these other published works and their findings.

Reply

As explained in our reply to Comment #4, an optimal search across *arbitrary* pulse energy modulations has not been achieved by previous deep learning-based simulators, or by other interpolation methods such as the Gaussian process regression approach taken in [Bamoto K, et al (2022)] [newly referenced as #53] for that matter.

The DL application mentioned in the review of [Kobayashi, et al (2021)] is the feature extraction also reported in Tani S, et al (2020), Optics Express 28(18), 26180-26188U, and the difference between this work and ours has been addressed in our reply to Comment #4.

[Bamoto K, et al (2022)] has been newly referenced as Ref #53 and mentioned in the manuscript as below.

Changes

The reference has been mentioned in the highlighted position:

Lines #172-173

...which has not been achieved by conventional deep learning-based schemes as well as other interpolation methods^{26,52,53} applied to laser processing.

Comment #6

There looks to be plenty of distinction in results, presentation, analysis etc. between these papers, but not in the DL/DNN methods used. A paper on deep learning applied to the laser welding process has been published [Gunther J, et al (2016) Mechatronics 34, 1–11]; the authors should include this in their review of literature and, if appropriate, qualify how their work builds on advances already made in applying DL to the broader discipline of laser processing.

Reply

Thank you for the reference. As the papers mentioned in Comment #3, the DL problem treated in Gunther et al [newly added as Ref #49] is different from ours which is the construction of a simulator for optimization. In Gunther et al, a DNN that extracts features and evaluates quality when given an image of the welded area was constructed, for a component of a real-time intelligent laser welding system. We have added this reference to the newly added sentence introducing the different kinds of DL applications in laser processing.

Change

The reference was added to the highlighted positions:

Lines #97-99

Due to this advantage, deep learning has been applied to various tasks in laser processing such as the prediction

of processing results^{27,28,47}, feature extraction^{49,50}, quality evaluation⁴⁹, and fetching the used processing parameters⁵¹.

Comment #7

The paper should be of strong interest to others in the field. The development of deep learning models/simulators is a good fit for manufacturing, since processes generate large amounts of data from machine sensors, yet currently most firms do not utilise this data effectively. Leading analysts forecast a trillion-plus USD market for deep learning models in both manufacturing and its wider sphere of supply chain management and logistics. There is increasing demand for such cyber-physical systems to enable new capability for: predictive maintenance to prevent in-production failures; predictive analytics for process optimisation; eliminating bottlenecks in the production cycle (e.g. materials supply); product development; quality assurance; and robotics, to handle complicated and hazardous processes / materials.

Reply:

Thank you for the comment.

Comment #8

Will the paper influence thinking in the field?

The model/simulator studied only considers 2 parameters, yet the laser drilling process using USPLs is known to be dependent on several applied parameters and conditions. Hence, a concern is the true novelty of what is achieved so far, and (realistically) how the approach could be viably translated to multiple parameters, more representative of a USPL drilling application. That said, the achievements being reported are certainly an important step in this direction.

Reply

Thank you for the comment. We would like to clarify the novelty of what we have achieved in our work, which we conveyed insufficiently in our previous manuscript.

It is true that our model was trained on energy-constant conditions which can be described just by 2 parameters: the pulse energy and the number of pulses. However, the actual optimization was conducted across energy-modulated conditions, which means that the energy of the pulse is no longer fixed throughout the drilling process, and a condition can no longer be described just by two parameters. For example, if we allow the pulse energy to be modulated up to N times (or modulation steps), the energy-modulated condition has N degrees of freedom, and the optimization problem becomes N -dimensional. The novelty of our achievement lies in the fact that we were able to run an optimization across a much higher-dimensional search space than the 2-parameter space consisting of energy-constant conditions. This was made possible by our iterative deep-learning scheme.

With our simulation method, we can run optimization problems with dimensions as high as $N=100$ in principle, since our simulator can be used to simulate drilling process of more than 1000 pulses, and a different pulse energy value can be inputted into the simulator every 10 pulses. However, since a 100-dimensional optimal search would

take too much time and would also be difficult to visualize, we chose a lower value of $N = 6$ in our work, to demonstrate the feasibility of our method.

As for the extension of our simulator to larger numbers of input processing parameters (such as pulse-width or spatial beam profile) is indeed an interesting and important topic that will be addressed in future works.

In order to sufficiently convey the novelty of what we have achieved using our simulator, we have added sentences and a graph covering what we have explained above. The modifications are listed in our reply to Comment #4, which is also related to the novelty of our work.

Comment #9

The authors state that the vast majority of previous models have not been validated outside the range of their experimental tuning parameters, and thus, their applicability for optimization is questionable. This could also be said of the case being reported here.

Reply

Thank you for the comment. We agree that the statement “the majority of previous models have not been validated outside the range of their experimental tuning parameters” was misleading, since it implies that the validity of our simulator had exceeded the trained parameter range.

What we meant to convey was that most previous models were only validated within the same parameter space from which the fitting data was collected and cannot be applied to optimization across a higher-dimensional search space. This would have been the case for our work *if* our model was only valid for energy-constant conditions within a certain range of pulse energy and numbers of pulses, in which case we could only explore a 2-dimensional parameter space which is the same as that from which the training data was collected. Instead, our simulator has been validated and applied across a higher-dimensional space than the 2-dimensional space consisting of energy-constant conditions. We have added Fig. 1a to the manuscript to better convey the difference between energy-constant conditions and energy-modulated conditions, which may help clarify our explanation here.

In order to avoid misunderstanding, we have made the modifications listed in our reply to Comment #4.

Comment #9

A challenge for deep learning in such a process is the large amount of data needed for accurate training of the model/simulator, with each item of data ‘handled’ requiring a process parameter to be changed (thus modifying the laser-material interaction) and an associated measurement to quantify the result and its level of success against chosen criteria. Collecting such data could take weeks; hence, not feasible. ‘Data augmentation’ methods are needed e.g. of captured images. For the claims to be convincing, there should at least be appropriate discussion to show an awareness of this challenge.

Reply

Thank you for the insightful comment. The mass preparation of training data is indeed a key ingredient for successful deep learning application. In our work, we were able to achieve rapid preparation of sufficient amounts of data due to the in-situ collection of microchannel images using a synchronized camera. Moreover, we augmented the training data by adding “zero-energy” data; therefore, we strongly agree that data augmentation is a powerful strategy for rapid data preparation. We have added the comment below in the “Training specifications of the DNN simulator” section of the Methods, emphasizing the importance of rapid and efficient training data preparation, and also how our augmentation procedure aligns with this.

Change

Lines #378-381

The rapid and efficient preparation of training data is a key ingredient for an effective deep learning application, and data augmentation methods such as the one conducted in our work are powerful techniques for achieving this, especially when there is a limit to how rapidly one can collect data by experimental means.

Comment #10

The authors claim that the simulator discovered a modulation condition 20% more energy-efficient than any from experimental data supplied for simulator training. From knowledge of laser drilling in both non-thermal (USPL) and thermal regimes, it seems intuitive and actually not surprising that a temporal variation in pulse energy of the form shown in video #4 gives the most effective result (increasing from an initial energy value, then falling away as a ‘tail’). However, from my understanding, it is not clear if the simulator was trained to seek out such a form. What was the decision-making applied at the start and during the process?

Reply

Thank you for the insightful question. We would like to clarify that the optimization that we conducted in our work was merely a “grid search-like” optimization, where all conditions were first simulated independently, after which the simulation results were evaluated. Therefore, no “active” decisions or adjustments were applied during each single drilling simulation to minimize the total pulse energy. However, we strongly agree that the development and employment of a more complex and time-efficient algorithm is of high interest, since even a virtual grid search becomes less feasible with the expansion of the search space, due to computational limits. We have added a comment on this topic in the Discussions section.

Changes

Lines #265-269

Although a grid search was employed in this study to demonstrate the applicability of our simulator to high-dimensional optimization, the employment of a more time-efficient algorithm is of high interest, since even a virtual grid search becomes less feasible with the expansion of the search space, due to computational limits.

Comment #11

How does the model respond to anomalies or an erroneous change in one laser parameter e.g. applied fluence?

What is its sensitivity to laser parameter fluctuations? This would seem important for future application to high precision ultrashort pulse machining involving highly nonlinear interaction mechanisms under industrial / engineering settings and conditions.

Reply

Thank you for the comment. We agree that the effect of processing parameter fluctuations on the final processing result is an industrially significant topic. Using our simulator, it is possible to rapidly determine such processing windows by running multiple simulations with slightly different input processing conditions, on top of rapid optimal search which was emphasized most in this work. We have added a comment in the Discussion section commenting on an additional way our simulator can contribute to the field of laser processing.

Change

Lines #271-273

The application of our simulator also extends to other important tasks in laser processing, such as rapidly investigating the effect of processing parameter fluctuations on the final processing result to determine processing windows.

Comment #12

I cannot offer an opinion on whether there are other experiments that would strengthen the paper further, how much would they improve it, and how difficult are they likely to be. However, addressing the points already made should help negate any need for further experimental work. There already appear to be ample results worthy of publication.

Overall, the claims appear to be appropriately discussed in the context of previous literature.

My opinion is that the manuscript is acceptable in its present form, taking into consideration the above comments and addressing concerns raised therein.

The manuscript is clearly written and appears to follow the required format. However, the current title is considered too broad to align with the claims of its findings. For example, a more appropriate title might be: 'A deep learning-based predictive simulator for 2-parameter optimization of ultrashort pulse laser drilling'.

Reply

Thank you for the suggestion. We agree that our previous title could be considered too broad and would like to propose the following alternative.

'A deep learning-based predictive simulator for high-dimensional optimization of ultrashort pulse laser drilling'

As we explained in our preceding replies, we believe the novelty of our simulator lies in its applicability to higher-dimensional optimization problems, more so than the number of parameters or range of parameter values. Although we chose a 6-dimensional optimization problem in this work to demonstrate the feasibility of our simulation method, we can run much higher-dimensional optimization problems in principle.

Comment #13

Regarding whether the manuscript could be shortened to aid communication of the most important findings, it is not clear to me how this could be achieved without removing important detail.

With some clarification to address the points raised, the authors' claims made can be justified and the article would not be 'overselling' these.

Overall, the authors appear to have been fair in their treatment of previous literature and this looks well-balanced for a full research article (compared to a review article). Some readers may benefit from including a little more insight into how the DL approach fits within – but is distinct from – the wider 'set' of machine learning (ML) and artificial intelligence (AI). An example could be the set diagram used by previous authors [see Mills, B., Grant - Jacob, J.A.:

Lasers that learn: the interface of laser machining and machine learning. IET Optoelectron. 15(5), 207–224 (2021). <https://doi.org/10.1049/ote2.12039>].

Reply

Thank you for the suggestion. In order to provide insight into how deep learning is distinct from its machine learning counterparts, we have added the sentences below.

Changes

Lines #92-99

Deep learning⁴⁸ is a subfield of machine learning, where a multi-layer function called the deep neural network is used to approximate input-output relationships. Its advantage over other machine learning methods is that the DNN can be designed and trained to directly approximate relationships between high-dimensional data such as images, which can be attributed to the massive amount of fitting parameters composing the DNN. Due to this advantage, deep learning has been applied to various tasks in laser processing such as the prediction of processing results^{27,28,47}, feature extraction^{49,50}, quality evaluation⁴⁹, and fetching the used processing parameters⁵¹.

Comment #14

Setting the work and its findings in the wider context of intelligent laser processing, there have been some relevant studies reported in additive manufacturing (specifically the selective laser melting or laser powder bed fusion process) which could be reviewed and mentioned if/as appropriate [Jayasinghe S, et al (2020), 'Automatic quality assessments of laser powder bed fusion builds from photodiode sensor measurements', Progress in Additive Manufacturing (2022) 7:143–160 <https://doi.org/10.1007/s40964-021-00219-w>]

Reply

Thank you for pointing out this reference. The machine learning method used in this reference, Gaussian process regression, is a similar method to deep learning in that they are both frequently used as an interpolation method in the field of laser processing, which is in good contrast with our deep learning-based scheme used to explore a higher-dimensional space. Therefore, we would like to add this to the references as an example of a different

machine learning algorithm applied to laser processing. The reference number is #52, and it has been referenced as below:

Changes

The reference was added to the highlighted position:

Lines #172-173

which has not been achieved by conventional deep learning-based schemes as well as other interpolation methods^{26,52,53} applied to laser processing.

Comment #15

Have the authors provided sufficient methodological detail that the experiments could be reproduced?

See comment below.

The statistical analysis of the data appears to be sound.

Should the authors be asked to provide further data or methodological information to help others replicate their work? (Such data might include source code for modelling studies, detailed protocols or mathematical derivations).

Detail of code would not be of interest / relevance here, but the operations carried out on the data should be clear. Fig. 1 already seeks to explain this. Perhaps a decision tree / flow diagram that could convey this in general terms for the journal readers, plus some brief detail of the numerical operations. It seems that more of this kind of detail was included in the authors' other recent publications on the topic.

Reply

Thank you for the suggestion. We have added a flow diagram explaining the entire process of our experiment, starting from the collection of the training data to the final grid-search optimization (Supplementary Fig. 8). Moreover, we have added the sentence below in the introduction, referring to Supplementary Fig. 8

Changes

Lines #83-84:

(See Supplementary Fig. 8 for an overview of the operations conducted in our work)

We look forward to hearing from you in due time regarding our submission and to respond to any further questions and comments you may have.

Sincerely,
Kohei Shimahara

REVIEWERS' COMMENTS:

Reviewer #1 (Remarks to the Author):

I am happy the reviewer comments have been answered and recommend the manuscript for publication.

Reviewer #2 (Remarks to the Author):

Dear authors, thank you very much for working out the text. Most of the former issues have been addressed with enough quality.

I have no other questions and recommended the article for the publication.

Reviewer #3 (Remarks to the Author):

The authors, in their rebuttal document, have given due consideration to all the points made in my previous review comments. The additional clarifications and explanations of various details are comprehensive and satisfactory. They would appear to reinforce the major claims of the paper, those being: demonstration of simulator-based optimization in ultrashort pulse laser drilling of glass, applying a deep learning (DL) approach; and the discovery by the simulator of a modulation condition 20% more energy-efficient than any from experimental data supplied for simulator training. The successful validation of the optimal condition found verifies the predictive capability potential of the deep learning-based approach for future smart laser processing systems.

I have reviewed the revised manuscript and can see that the authors have incorporated some additional text, additional references and a diagram in agreement with my recommendations. I have not noticed any errors or identified any additional items that require attention. There appears to be sufficient detail for the work to be reproducible by others. I am therefore recommending to the editors that, in my view and based on my knowledge and expertise, the manuscript should be published.